# Seeds, Contexts, and Tongues: Decoding the Drivers of Hallucination in Language Models

## Abstract

This study investigates hallucinations in Large Language Models (LLMs), particularly in Nigerian and Western contexts. We study how hyperparameters, cultural background, and prompt language (particularly, Nigerian Pidgin) affect hallucination rates. Using semantic entropy as an indicator of hallucination, we examine response variability in Llama 3.1 outputs and cluster them using the entailment model microsoft/deberta-base-mnli to identify semantic similarity. We then use these clusters to calculate semantic entropy (the variation in meanings of the LLM's responses) using a variant of Shannon entropy to quantify hallucination likelihood. Our findings shed light on ways to improve LLM reliability and consistency across linguistic and cultural situations.

**Keywords:** Language Models (LLMs), Hallucination Detection, Semantic Entropy, Natural Language Processing (NLP), Pidgin Language, Cross-Lingual Analysis, Hyperparameters

## 1  Introduction

Large Language Models (LLMs) like ChatGPT, Gemini, and Claude are changing how people find and interact with information. Many users now prefer them over traditional search engines. However, a major challenge remains: hallucinations—when these models generate responses that sound correct but are actually false or misleading. Researchers have defined hallucinations in different ways. Rawte et al. (2023) describes them as instances where an LLM produces information that deviates from reality or includes fabricated content. Others, like Huang et al. (2024), classify hallucinations into two types: factual and faithful. Factual hallucinations happen when a response contradicts real-world facts, while faithfulness hallucinations occur when an answer strays from the user's intent. How we define hallucinations shapes how we try to fix them.

Measuring hallucinations is tricky, especially with open-ended questions. Unlike yes/no questions that test memory, open-ended ones assess reasoning and decision-making—areas where LLMs still struggle. But comparing an LLM's response to a "ground truth" in these cases isn't straightforward. To tackle this, Duan et al. (2024) explored uncertainty-based methods using LLMs' hidden states to detect hallucinations. But these techniques don't always work, especially with closed-source models. Inspired by Farquhar et al. (2024), who used semantic entropy to detect hallucinations, we take this a step further. We examine how hyperparameters, prompt language (especially Nigerian Pidgin), and cultural context affect semantic entropy in LLM responses. Our goal is to understand how these factors influence hallucination rates in Nigerian and Western settings. By doing this, we hope to offer practical insights into making LLMs more reliable across different languages and cultures.

## 2  Related Works

Hallucinations in Large Language Models (LLMs) have become a hot topic as these models play an increasingly central role in how people access and generate information. Although LLMs like ChatGPT, Gemini, and Claude can produce impressively fluent responses, they sometimes generate content that sounds convincing yet is factually incorrect or misleading. Researchers have been

working hard to understand why this happens and how to reduce it. Hallucination is touted to be introduced to LLMs through flaws in data, training and inference. Issues like misinformation and biases, knowledge shortcut and knowledge recall failures, architecture flaws and suboptimal training objectives, capability misalignment and belief misalignment are the go-to factors. Some researchers argue that hallucinations should be considered as a natural component of the generating process rather than a flaw to be fixed. According to Rawte et al.(2023), the diversity of LLM outputs, commonly referred to as hallucinations, may indicate the model's creativity. However, this trait is not applicable in every case. As a result, it is crucial to understand what circumstances affect it the most and how to control them.

One common way to frame the problem is by looking at two key aspects: **factual accuracy** and **faithfulness to the input**. For example, Rawte et al. (2023) describes hallucinations as moments when a model's output deviates from reality or even invents information. In a similar vein, Huang et al. (2024) splits hallucinations into two types: **factual hallucinations**, where responses conflict with known facts, and **faithfulness hallucinations**, where the answer does not properly reflect the question's intent. These distinctions are important because they suggest different strategies for reducing errors. Traditionally, researchers have tackled hallucinations in big language models by directly comparing the generated outputs to some predetermined ground truth. This strategy is effective for jobs with clear, organized solutions, such as factual question-answering, but it falls short when used to more open-ended tasks, such as summarizing, creative writing, or reasoning. In these situations, because there is no single "correct" solution, focusing simply on ground truth may oversimplify the situation and overlook more subtle sorts of hallucinations.

To address these issues, some academics have turned to uncertainty estimation as a method of detecting hallucinations. For instance, Duan et al. (2024) proposes leveraging language models' hidden states as inputs to regression models that assess uncertainty. The idea is that the model's internal representations might reveal how confident it is in its output. However, this method might be troublesome in black-box models because these concealed states are difficult to obtain. As a solution, other studies have investigated using simpler indicators as proxies for uncertainty, such as answer length or distance between embeddings in a semantic space. The underlying idea is that an unusually extended or semantically distant response to the prompt could indicate that the model is "making things up." However, these methods are not foolproof; for example, a lengthy response does not always indicate a mistake, and embedding-based measurements frequently rely on valid reference points.

Overall, uncertainty estimating techniques, particularly those based on regression or classification, perform best in closed-ended tasks with clear correct or erroneous outputs. Classification models are frequently developed to distinguish between true and false responses using annotated data. However, for free-form text generation, when there is no one "right" response and the evaluation must reflect more complex thinking, standard systems face considerable challenges.

To fill this gap, Farquhar et al. (2024) introduced a method with semantic grounding. This involved using semantic entropy, which evaluates the randomness of a model's replies. The idea is simple: if a model is uncertain, it will offer a variety of responses with different meanings when given the same prompt multiple times. As a result, high semantic entropy can be used to detect hallucinations in the model. Unlike regression-based approaches, which rely on hand-crafted features, semantic entropy provides a more adaptable and model-agnostic approach that works even when the model's inner workings are obscured.

Our research extends these theories by looking into how hyperparameter settings, language, and cultural context influence hallucination detection. We focus on Nigerian Pidgin, a language with its distinct structure and idiosyncrasies, to examine how existing uncertainty-based algorithms work when applied to a low-resource language. By adding these linguistic and cultural aspects, we want to improve the robustness and usefulness of uncertainty estimating strategies for detecting hallucinations across a broader range of languages and real-world circumstances.

# 3 METHODOLOGY

## 3.1 DATASETS

We used two datasets, each consisting of 20 questions:

- **Western-based questions:** The first twenty questions from TriviaQA (Joshi et al., 2017).

- **Nigerian-based questions:** A set of questions designed to be culturally and contextually relevant to Nigeria, mirroring the domains found in the Western dataset.

The questions cover a variety of domains, including literature and arts, history, geography, music and entertainment, religion and culture, law and governance, sports, science and technology, economics and resources, and miscellaneous trivia. The average question length is 51.95 characters.

## 3.2 EXPERIMENTAL DESIGN

We conducted several experiments with the Large Language Model (LLM) to observe its behavior under different conditions. In the first batch of experiments, we queried the LLM with questions from the Western-based dataset and then the Nigerian-based one in Nigerian Pidgin under certain conditions. Each question in the dataset was asked multiple times in each experiment. The conditions were a varied random seed while keeping the temperature constant at 0 and 1 respectively in different experimental runs. The aim was to investigate the impact of the random seed on semantic entropy and, by extension, hallucination, when an LLM is queried in Nigerian Pidgin at both temperature extremes. In the second batch of experiments, we queried the LLM in English with questions from the Western-based dataset and then the Nigerian-based one, under similar conditions. This aimed to investigate the impact of the random seed on semantic entropy and, by extension, hallucination, when an LLM is queried in English at both temperature extremes. We used the Spearman rank correlation to figure out the relationship between our control variable and semantic entropy in the first and second batches of experiments. This was important to determine how well the rankings of the random seed and semantic entropy match, even if the connection is not linear. This is great for our data, as the relationship between variables may not be a clear, straight-line trend. Finally, in the third experiment, to isolate the effect of prompt language (English and Nigerian Pidgin). We varied the prompt language while keeping other variables constant at two different temperature settings: 0 and 1. Here, we used the point-biserial correlation instead of Spearman rank. This is because we represented the prompt language with categorical variables and that particular type of correlation assesses the strength and direction of the association between one continuous variable and one binary categorical variable. These experiments allowed us to explore how different factors, such as temperature, language, and random seed, influence the model's output.

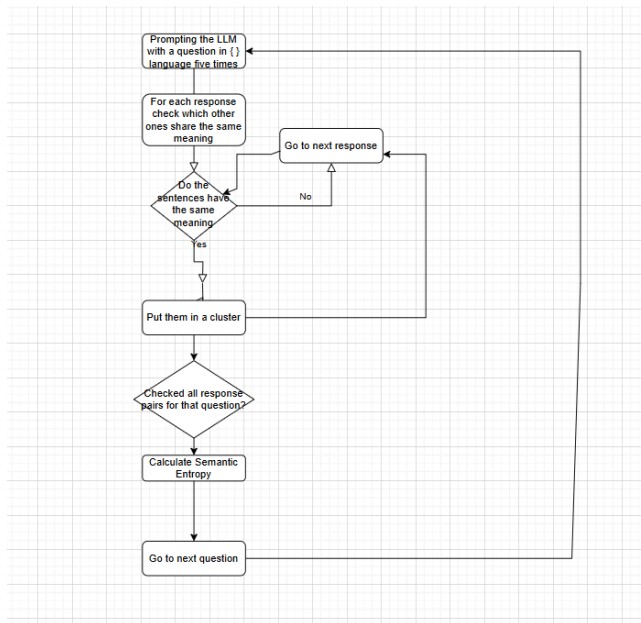

Figure 1: Experimental Design Overview

### 3.3 SEMANTIC ENTROPY COMPUTATION

Semantic entropy quantifies uncertainty in LLM responses:

$$H = -\sum_{i=1}^{N} p_i \log p_i \qquad (1)$$

where $N$ is the number of meaning clusters, and $p_i$ is the probability of the $i$-th meaning. Higher entropy indicates more varied responses, suggesting increased hallucination risk.

## 4 RESULTS

### 4.1 EFFECT OF SEED ON SEMANTIC ENTROPY (PIDGIN PROMPTS)

**Western-based dataset (T = 1.0, Pidgin prompts):**
Here, we prompted the LLM in Pidgin using questions in the Western-based dataset at T=1. After doing so, we observed a moderate negative Spearman correlation ($r = -0.600, p = 0.005$) between the seed and semantic entropy. This suggests that as the seed value increases, the semantic entropy tends to decrease, indicating that higher seeds consistently produce responses with lower variability. Conversely, lower seed values tend to generate more varied or uncertain responses. **Western-based dataset (T = 0.1, Pidgin prompts):**
Here, we prompted the LLM in Pidgin from questions in the Western-based dataset at T=0.1. After doing so, a weak positive correlation ($r = 0.204, p = 0.388$) was observed, suggesting that sometimes as seed increases the semantic entropy increases. However, the p-value is quite high meaning that this emergent relationship is very likely down to chance. So at low temperatures changing the seed doesn't significantly increase hallucination even in Pidgin.

**Nigerian-based dataset (T = 0.1, Pidgin prompts):**
Here, we prompted the LLM in Pidgin from questions in the Nigerian-based dataset at T=0.1. After doing so, a moderate positive Spearman correlation ($r = 0.504, p = 0.002$) was observed suggesting that seed changes have a non-trivial effect on entropy at low temperatures. So, unlike the Western dataset where no relationship emerges between seed and semantic entropy at low temperatures, we have the opposite here. When prompting LLMs with questions from the Nigerian context at low temperatures, increasing the seed can lead to semantic entropy rising.

Table 1: Western Context: Varying Seed in Pidgin

(a) Temp = 1.0

| Sem Ent | Seed | Temp | Lang | Context |
|---------|------|------|------|---------|
| 0.00 | 3197 | 1.0 | 0 | Western |
| 2.32 | 18 | 1.0 | 0 | Western |
| 0.46 | 3372 | 1.0 | 0 | Western |
| 1.37 | 1654 | 1.0 | 0 | Western |
| 0.53 | 1029 | 1.0 | 0 | Western |
| 0.97 | 1997 | 1.0 | 0 | Western |
| 0.72 | 191 | 1.0 | 0 | Western |
| 0.00 | 4675 | 1.0 | 0 | Western |
| 0.00 | 3117 | 1.0 | 0 | Western |
| 1.44 | 3389 | 1.0 | 0 | Western |
| 1.23 | 213 | 1.0 | 0 | Western |
| 2.32 | 1202 | 1.0 | 0 | Western |
| 2.32 | 2122 | 1.0 | 0 | Western |
| 0.00 | 3834 | 1.0 | 0 | Western |
| 0.00 | 4737 | 1.0 | 0 | Western |
| 0.46 | 1617 | 1.0 | 0 | Western |
| 0.53 | 4686 | 1.0 | 0 | Western |
| 1.19 | 2921 | 1.0 | 0 | Western |
| 0.00 | 4410 | 1.0 | 0 | Western |
| 1.41 | 2215 | 1.0 | 0 | Western |

(b) Temp = 0.1

| Sem Ent | Seed | Temp | Lang | Context |
|---------|------|------|------|---------|
| 0.00 | 4473 | 0.1 | 0 | Western |
| 2.32 | 1400 | 0.1 | 0 | Western |
| 0.00 | 541 | 0.1 | 0 | Western |
| 0.72 | 686 | 0.1 | 0 | Western |
| 0.00 | 1288 | 0.1 | 0 | Western |
| 0.00 | 2323 | 0.1 | 0 | Western |
| 0.00 | 2028 | 0.1 | 0 | Western |
| 0.00 | 4355 | 0.1 | 0 | Western |
| 0.00 | 2022 | 0.1 | 0 | Western |
| 0.00 | 1040 | 0.1 | 0 | Western |
| 0.72 | 4791 | 0.1 | 0 | Western |
| 2.32 | 3838 | 0.1 | 0 | Western |
| 0.00 | 3166 | 0.1 | 0 | Western |
| 0.26 | 4570 | 0.1 | 0 | Western |
| 0.00 | 705 | 0.1 | 0 | Western |
| 0.26 | 4144 | 0.1 | 0 | Western |
| 0.00 | 1489 | 0.1 | 0 | Western |
| 0.00 | 2221 | 0.1 | 0 | Western |
| 0.00 | 3473 | 0.1 | 0 | Western |
| 0.00 | 3986 | 0.1 | 0 | Western |

Nigerian-based dataset (T = 1.0, Pidgin prompts) Here, we prompted the LLM in Pidgin using questions in the Nigerian-based dataset at T=1.0. After doing so, a moderate positive Spearman correlation ($r = 0.29$, $p = 0.2$) was observed. Despite the discovered positive relationship, the effect of seed alterations on entropy was minor and statistically negligible. At high temperatures, the LLM's output fluctuation is already large, thus seed modifications have little impact. This contradicts our findings from the Western dataset, which indicated that seed changes had a bigger influence at lower temperatures.

Table 2: Nigerian context in Pidgin (Temperature = 0.1)

| Sem. Ent. | Seed | Temp | Lang | Context |
|-----------|------|------|------|---------|
| 0.44 | 3534 | 0.1 | 0 | Nigerian |
| 1.16 | 4040 | 0.1 | 0 | Nigerian |
| 1.50 | 4822 | 0.1 | 0 | Nigerian |
| 0.97 | 920 | 0.1 | 0 | Nigerian |
| 0.44 | 3493 | 0.1 | 0 | Nigerian |
| 2.32 | 3347 | 0.1 | 0 | Nigerian |
| 0.00 | 1076 | 0.1 | 0 | Nigerian |
| 0.72 | 1026 | 0.1 | 0 | Nigerian |
| 2.32 | 2571 | 0.1 | 0 | Nigerian |
| 2.32 | 4670 | 0.1 | 0 | Nigerian |
| 1.37 | 2701 | 0.1 | 0 | Nigerian |
| 0.97 | 205 | 0.1 | 0 | Nigerian |
| 2.32 | 2015 | 0.1 | 0 | Nigerian |
| 0.00 | 2105 | 0.1 | 0 | Nigerian |
| 0.00 | 3327 | 0.1 | 0 | Nigerian |
| 0.00 | 385 | 0.1 | 0 | Nigerian |
| 0.00 | 614 | 0.1 | 0 | Nigerian |
| 0.97 | 2493 | 0.1 | 0 | Nigerian |
| 2.32 | 4590 | 0.1 | 0 | Nigerian |
| 1.52 | 3635 | 0.1 | 0 | Nigerian |

Table 3: Nigerian context in Pidgin (Temperature = 1.0)

| Sem. Ent. | Seed | Temp | Lang | Context |
|-----------|------|------|------|---------|
| 0.00 | 2614 | 1.0 | 0 | Nigerian |
| 1.92 | 1782 | 1.0 | 0 | Nigerian |
| 0.72 | 4089 | 1.0 | 0 | Nigerian |
| 1.44 | 587 | 1.0 | 0 | Nigerian |
| 0.97 | 2917 | 1.0 | 0 | Nigerian |
| 2.32 | 2985 | 1.0 | 0 | Nigerian |
| 0.44 | 1858 | 1.0 | 0 | Nigerian |
| 0.00 | 4452 | 1.0 | 0 | Nigerian |
| 0.26 | 179 | 1.0 | 0 | Nigerian |
| 2.32 | 3535 | 1.0 | 0 | Nigerian |
| 1.92 | 3354 | 1.0 | 0 | Nigerian |
| 0.26 | 344 | 1.0 | 0 | Nigerian |
| 1.37 | 447 | 1.0 | 0 | Nigerian |
| 0.00 | 2890 | 1.0 | 0 | Nigerian |
| 2.45 | 4680 | 1.0 | 0 | Nigerian |
| 0.79 | 102 | 1.0 | 0 | Nigerian |
| 0.99 | 3993 | 1.0 | 0 | Nigerian |
| 0.97 | 570 | 1.0 | 0 | Nigerian |
| 0.97 | 381 | 1.0 | 0 | Nigerian |
| 1.44 | 3221 | 1.0 | 0 | Nigerian |

## 4.2 EFFECT OF SEED ON SEMANTIC ENTROPY (ENGLISH PROMPTS)

In an attempt to isolate the effect of the prompt language on hallucination by treating it like another variable. So we ask the LLM questions from the Western-based dataset and Nigerian-based dataset in different experimental runs respectively to see if there are differences in LLM behaviour between contexts. We represent the prompt languages with categorical variables (Nigerian Pidgin:0 and English:1).

**Western-based dataset:** We asked the LLM in English questions from the Western dataset at $T = 0.1$. A weak negative Spearman correlation ($r = -0.09$, $p = 0.71$) was discovered, indicating that the relationship between seed and semantic entropy is most likely due to random chance. This finding is congruent with the findings of prompting in Nigerian Pidgin at low temperatures. At $T = 1.0$, a weak positive relationship ($r = 0.23$, p = 0.33) was discovered. Although this result is statistically more significant than the previous one, the effect remains minor and is most likely due to random chance. When asking LLMs in English with Western-context questions, modifying the seed does not appear to have a substantial impact on semantic entropy or hallucinations, demonstrating that seed variants are ineffective in this environment.

Table 4: Western Context in English (Temperature = 0.1)

| Sem Ent | Seed | Temp | Lang | Context |
|---|---|---|---|---|
| 0.0 | 4695 | 0.1 | 1 | Western |
| 0.0 | 2878 | 0.1 | 1 | Western |
| 2.3219 | 2839 | 0.1 | 1 | Western |
| 0.0 | 550 | 0.1 | 1 | Western |
| 2.3219 | 2222 | 0.1 | 1 | Western |
| 0.0 | 3187 | 0.1 | 1 | Western |
| 2.3219 | 1445 | 0.1 | 1 | Western |
| 2.3219 | 3739 | 0.1 | 1 | Western |
| 0.0 | 1330 | 0.1 | 1 | Western |
| 2.3219 | 456 | 0.1 | 1 | Western |
| 0.0 | 3922 | 0.1 | 1 | Western |
| 2.3219 | 2547 | 0.1 | 1 | Western |
| 2.3219 | 332 | 0.1 | 1 | Western |
| 2.3219 | 3903 | 0.1 | 1 | Western |
| 2.3219 | 3612 | 0.1 | 1 | Western |
| 0.0 | 696 | 0.1 | 1 | Western |
| 2.3219 | 4137 | 0.1 | 1 | Western |
| 2.3219 | 424 | 0.1 | 1 | Western |
| 0.0 | 1409 | 0.1 | 1 | Western |
| 0.4422 | 4543 | 0.1 | 1 | Western |

Table 5: Western Context in English (Temperature = 1.0)

| Sem Ent | Seed | Temp | Lang | Context |
|---|---|---|---|---|
| 0.44218 | 45 | 1.0 | 1 | Western |
| 0.0 | 3410 | 1.0 | 1 | Western |
| 2.32193 | 4744 | 1.0 | 1 | Western |
| 0.0 | 1332 | 1.0 | 1 | Western |
| 2.32193 | 3368 | 1.0 | 1 | Western |
| 0.52871 | 127 | 1.0 | 1 | Western |
| 0.0 | 4615 | 1.0 | 1 | Western |
| 0.44218 | 1211 | 1.0 | 1 | Western |
| 0.46439 | 2925 | 1.0 | 1 | Western |
| 2.32193 | 639 | 1.0 | 1 | Western |
| 1.52193 | 2643 | 1.0 | 1 | Western |
| 2.32193 | 3586 | 1.0 | 1 | Western |
| 2.32193 | 682 | 1.0 | 1 | Western |
| 0.72193 | 2572 | 1.0 | 1 | Western |
| 0.0 | 2185 | 1.0 | 1 | Western |
| 0.0 | 1074 | 1.0 | 1 | Western |
| 0.0 | 1092 | 1.0 | 1 | Western |
| 2.32193 | 3619 | 1.0 | 1 | Western |
| 0.0 | 530 | 1.0 | 1 | Western |
| 0.44218 | 1634 | 1.0 | 1 | Western |

**Nigerian-based dataset:** We asked the LLM in English questions from the Nigerian-based dataset at $T = 0.1$, and a weak negative correlation ($r = -0.01$, $p = 0.96$) was observed indicating a trivial relationship between both variables.

At $T = 1$, a weak negative correlation ($r = -0.26$, $p = 0.27$) was observed indicating a trivial relationship between both variables.

## 4.3 EFFECT OF PROMPT LANGUAGE ON SEMANTIC ENTROPY

## 4.4 WESTERN-BASED DATASET

At $T = 0.1$, a weak positive correlation ($r = 0.218$, $p = 0.177$) was observed. This suggests that the relationship between language and semantic entropy is likely due to random variation rather than any meaningful pattern. However, at $T = 1$, a moderate negative correlation ($r = -0.476$, $p = 0.002$) was found, indicating a more substantial relationship between the prompt language and

Table 6: Nigerian Context in English at Temperature 0.1

| Sem Ent | Seed | Temp | Lang | Context |
|---|---|---|---|---|
| 0.0 | 3331 | 0.1 | 1 | Nigerian |
| 0.0 | 2059 | 0.1 | 1 | Nigerian |
| 2.3219 | 613 | 0.1 | 1 | Nigerian |
| 0.0 | 1324 | 0.1 | 1 | Nigerian |
| 0.0 | 2530 | 0.1 | 1 | Nigerian |
| 0.0 | 3180 | 0.1 | 1 | Nigerian |
| 0.0 | 377 | 0.1 | 1 | Nigerian |
| 0.0 | 3207 | 0.1 | 1 | Nigerian |
| 2.3219 | 2973 | 0.1 | 1 | Nigerian |
| 0.0 | 4649 | 0.1 | 1 | Nigerian |
| 2.3219 | 4649 | 0.1 | 1 | Nigerian |
| 2.3219 | 2711 | 0.1 | 1 | Nigerian |
| 2.3219 | 4168 | 0.1 | 1 | Nigerian |
| 2.3219 | 2370 | 0.1 | 1 | Nigerian |
| 0.0 | 2129 | 0.1 | 1 | Nigerian |
| 0.0 | 608 | 0.1 | 1 | Nigerian |
| 0.0 | 199 | 0.1 | 1 | Nigerian |
| 0.9705 | 1011 | 0.1 | 1 | Nigerian |
| 0.0 | 2555 | 0.1 | 1 | Nigerian |

Table 7: Nigerian Context in English at Temperature 1.0

| Sem Ent | Seed | Temp | Lang | Context |
|---|---|---|---|---|
| 2.3219 | 2489 | 1.0 | 1 | Nigerian |
| 2.3219 | 2359 | 1.0 | 1 | Nigerian |
| 0.9710 | 901 | 1.0 | 1 | Nigerian |
| 0.7219 | 3047 | 1.0 | 1 | Nigerian |
| 0.0 | 1643 | 1.0 | 1 | Nigerian |
| 2.3219 | 3641 | 1.0 | 1 | Nigerian |
| 0.0 | 1474 | 1.0 | 1 | Nigerian |
| 0.0 | 4547 | 1.0 | 1 | Nigerian |
| 0.0 | 1897 | 1.0 | 1 | Nigerian |
| 2.3219 | 2209 | 1.0 | 1 | Nigerian |
| 1.3710 | 3417 | 1.0 | 1 | Nigerian |
| 2.3219 | 373 | 1.0 | 1 | Nigerian |
| 2.3219 | 2958 | 1.0 | 1 | Nigerian |
| 2.3219 | 916 | 1.0 | 1 | Nigerian |
| 0.0 | 3153 | 1.0 | 1 | Nigerian |
| 0.0 | 4838 | 1.0 | 1 | Nigerian |
| 1.4997 | 3610 | 1.0 | 1 | Nigerian |
| 0.2575 | 4029 | 1.0 | 1 | Nigerian |
| 0.0 | 3225 | 1.0 | 1 | Nigerian |
| 0.0 | 3030 | 1.0 | 1 | Nigerian |

entropy. This suggests that as temperature increases, the language used in the prompt has a more pronounced effect on semantic entropy.

Figure 2: Table 1: Lang. Infl. (T=0.1, Seed=c)

| Sem. Entropy | Temp. | Lang. | Context |
|---|---|---|---|
| 1.628 | 0.1 | 1 | Western |
| 1.371 | 0.1 | 1 | Western |
| 0.258 | 0.1 | 1 | Western |
| 1.186 | 0.1 | 1 | Western |
| 1.371 | 0.1 | 1 | Western |
| 0.258 | 0.1 | 0 | Western |
| 0.722 | 0.1 | 0 | Western |
| 2.322 | 0.1 | 0 | Western |
| 1.371 | 0.1 | 0 | Western |
| 0.464 | 0.1 | 1 | Western |
| 0.722 | 0.1 | 0 | Western |
| 1.922 | 0.1 | 0 | Western |
| 1.900 | 0.1 | 1 | Western |
| 1.628 | 0.1 | 1 | Western |
| 1.922 | 0.1 | 0 | Western |
| 0.613 | 0.1 | 1 | Western |
| 0.464 | 0.1 | 1 | Western |
| 1.186 | 0.1 | 1 | Western |
| 1.628 | 0.1 | 1 | Western |
| 0.722 | 0.1 | 0 | Western |

Figure 3: Table 2: Lang. Infl. (T=1.0)

| Sem. Entropy | Temp. | Lang. | Context |
|---|---|---|---|
| 2.386 | 1.0 | 0 | Western |
| 1.715 | 1.0 | 0 | Western |
| 2.364 | 1.0 | 0 | Western |
| 1.922 | 1.0 | 1 | Western |
| 0.000 | 1.0 | 1 | Western |
| 1.835 | 1.0 | 0 | Western |
| 2.322 | 1.0 | 1 | Western |
| 0.258 | 1.0 | 1 | Western |
| 2.322 | 1.0 | 0 | Western |
| 2.322 | 1.0 | 1 | Western |
| 1.922 | 1.0 | 1 | Western |
| 0.722 | 1.0 | 1 | Western |
| 0.722 | 1.0 | 1 | Western |
| 0.258 | 1.0 | 1 | Western |
| 1.371 | 1.0 | 1 | Western |
| 1.835 | 1.0 | 0 | Western |
| 1.779 | 1.0 | 0 | Western |
| 1.371 | 1.0 | 1 | Western |
| 1.715 | 1.0 | 0 | Western |
| 2.000 | 1.0 | 1 | Western |

## 4.5 NIGERIAN-BASED DATASET

At $T = 1$, the correlation between language and semantic entropy was negligible ($r \approx 0$, $p = 1.0$), implying that prompt language had no significant effect on entropy within this context. However, at $T = 0.1$, a weak negative correlation ($r \approx -0.212$, $p = 0.188$) was observed. Although this correlation was more significant than at higher temperatures, it was still not strong enough to rule out the possibility that the observed relationship was due to random chance.

## 5 CONCLUSION

Our findings indicate that seed selection, temperature, and cultural context affect hallucination rates differently across datasets. There seems to be a broadly negative correlation between semantic entropy and, seed and prompt language (at least the statistically significant ones) on the Western context. However, the relationship between semantic entropy and, seed and prompt language only seems to emerge at high temperatures and with Pidgin prompts in Nigerian context. Conversely, any relationship between semantic entropy and, seed and prompt language only seems to emerge at low temperatures and with Pidgin prompts. The relationship is however a moderate positive correlation. This means that to immprove model reliability when dealing with Western context and Pidgin prompts we make our temperature values as low as possible. Conversely, to achieve the same with Nigerian context and Pidgin prompts we make the temperature as high as possible. Pidgin prompts exhibit distinct entropy patterns, highlighting linguistic influences on model reliability. Further work could extend the analysis to additional African languages, improving LLM adaptation across diverse linguistic landscapes.

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
