# OpenReview forum: "Seeds, Contexts, and Tongues: Decoding the Drivers of Hallucination in Language Models"
_ICLR.cc/2025/Workshop/BuildingTrust — Submitted to BuildingTrust_

### Official Review · Reviewer_gLmi · 2025-02-18
**Interesting ideas, but requires further work**

**Rating:** 3
**Confidence:** 5

**Review:**

## Summary ##

This paper investigates the influence of hyperparameters (seed, and temperature) and prompt language on the estimated semantic entropy. Semantic entropy is a method introduced by Kuhn et al. (2023, https://arxiv.org/abs/2302.09664) that detects hallucinations in LLMs by computing the entropy over semantically-equivalent clusters.

Overall, I think that the research question studied is interesting, but the paper requires further work before it can be accepted. In particular, I think the experiment needs to be further developed (e.g., should include a larger sample size than 20 prompts) and the write-up needs to be improved (e.g., the wrong equation for semantic entropy is provided).

## Strengths ##

I think the topic chosen is interesting. The authors analyse the influence of hyperparameters (seed value, temperature) and prompt language on the estimated semantic entropy. Temperature was studied in the original semantic entropy paper (https://arxiv.org/abs/2302.09664). However, I think the investigation into seed and prompt language are interesting.
From a robustness perspective, it is important for semantic entropy to be robust to seed selection. Similarly, I think that research into the performance of hallucination detection methods on languages such as Nigerian Pigdin is important for the community. LLM research is often English-centric, and it is important that these methods perform well on non-English languages.

## Weaknesses ##

[1] The wrong equation for semantic entropy is provided in the paper. The authors provide the formula for entropy, which is meaningfully different. Semantic entropy estimates:
$$|C|^{-1} \sum \log p(C_i|x)$$
$|C|^{-1}$ doesn’t need to be the same as $p(C_i|x)$ — and mostly likely is not. As the code is not provided, I'm not sure which equation was implemented.

[2] Experiment:
- the experiment sample size is small (20 prompts), making it difficult to draw conclusions -- I would expect at least 100 prompts.
- the details provided are not enough to re-produce the experiments, e.g., 'the nigerian-based questions' dataset is private, which makes it difficult to understand the nature of the dataset. Examples of prompts would be useful.

[3] Presentation and discussion of the results
- the presentation of the results should be improved -- the main results focus on whether there's a correlation between semantic entropy and the studied. As such, a scatter plot (perhaps with a line showing the estimated correlation) or bar plots seems more appropriate.
- it would be great if the authors discussed the results more. One aspect I find difficult to wrap my head around is that the seed magnitude influences the semantic entropy estimation.  I think the authors should elaborate more on this result and perhaps run further experiments to explain the phenomenon.

[4] Writing -- the authors should be more careful with the language/claims, and the paper clarity could be improved.
One example where clarity could be improved is the introduction. Currently, it heavily focuses on hallucinations, and the method (and shortcomings) of duan et al. This part is repeated in the related work. However, duan et al., is not directly relevant to the work. Instead, I think it would be more appropriate to discuss multilingual LLM behaviour and robustness.
E.g., a possible structure for the introduction could be:
(1) introduce hallucinations,
(2) introduce hallucination detection methods,
(3) mention that methods predominantly focus on English text evaluation + mention inconsistencies in multi-lingual behaviour (e.g., the propensity of LLMs to answer in English, or assign higher probabilities to English tokens)
(4) introduce problems with robustness
(5) discuss paper contributions
(3) and (4) are currently missing from the paper, which would provide for a more natural transition into (5).

A couple of examples where the text could be improved are:
- “Unlike yes/no questions that test memory” -> I'd be careful, as this is an oversimplification of yes/no classification questions
- “Hallucination is touted to be introduced to LLMs through flaws in data, training and inference. Issues like misinformation and biases, knowledge shortcut and knowledge recall failures, architecture flaws and suboptimal training objectives, capability misalignment and belief misalignment are the go-to factors.” -> needs citations
- “the diversity of LLM outputs, commonly referred to as hallucinations, may indicate the model’s creativity.”  -> I briefly looked into the reference, and couldn't find this claim in Rawte et al. Rawte et al claim that ‘hallucinations’ are not harmful, and can be used in creative endeavours. However, I couldn't find that 'diversity of LLM outputs are commonly referred to as hallucinations'

Minor details:
- Use `` for the left quotation marks
- Contractions should be removed (don’t, isn’t, etc.)
- Citations are inconsistent (e.g., line 059) — should use \citet and \citep
- Line 069 big language models -> large language models
- Lines 080-083: missing citations
- Line 142: LLM -> Llama 3.1
- Figure 1: The legibility could be improved -- e.g., increase font size. It's missing part of what happens when a sentence is not part of a cluster: I.e., the step:
Does sentence have the same meaning as any of the existing clusters?
If no -> place sentence in new cluster

---

### Official Review · Reviewer_BcXu · 2025-02-27
**Interesting Problem but Incomplete Work**

**Rating:** 4
**Confidence:** 4

**Review:**

The authors aim to study the impact of random seeds, various cultural contexts, as well as the language of the prompt itself, on the eventual hallucinations.

The problem statements tackled and the setup itself are interesting. However, the work is clearly incomplete, and makes some absurd claims instead of exploring the problem deeper.

(a) The authors suggest that the seeds have some substantial negative correlation with semantic entropy, and thus claim that higher seeds would create lower entropy. Random seeds DO NOT have comparative value (if they do in the setup used by the authors, please clarify, because they are not supposed to). To extend the authors' claim, I ask, what would be the semantic entropy if the random seed was, say, 1e10. Similarly, what would be the semantic entropy if the random seed was, say, -1e10? Are the authors trying to say that we can reduce hallucinations by choosing a larger seed? I believe they are getting these absurd results because the experiment setup is quite small (only 20 questions from each language), but the lack of understanding of the claims being made is not good.

(b) Most experiment discussions are limited to just measuring the correlation, with no overarching conclusions/takeaways. What do we learn from these experiments? The only takeaway that is actually present is the one that claims correlation with the random seed, which is quite absurd to me.

(c) The paper writing itself is incomplete. There is an empty section 4.3.

The authors have a very impactful research question to explore, however, this is a very preliminary draft of something interesting to come. This is, unfortunately, not in a shape to be accepted.

---

### Decision · Program_Chairs · 2025-03-04

**Decision:**

Reject

**Comment:**

The paper's experimental setup is limited, with an extremely small sample size (only 20 prompts), making its conclusions unreliable, especially regarding the claim of a negative correlation between random seeds and semantic entropy. Additionally, the writing is incomplete and unclear, with missing sections, incorrect equations, and insufficient discussion of results, leaving fundamental claims unsubstantiated and raising concerns about the validity of the findings.